# Subgenomic RNA and Limited Cross-Reactive Neutralising Antibodies Point to Potential Improvements in SARS-CoV-2 Clinical Handling

**DOI:** 10.3390/ijms26072948

**Published:** 2025-03-24

**Authors:** Carlos Davina-Nunez, Sonia Perez-Castro, Jorge Julio Cabrera-Alvargonzalez, Elena Gonzalez-Alonso, Sergio Silva-Bea, Miriam Rodriguez-Perez, Maria del Pilar Figueroa-Lamas, Alexandre Perez-Gonzalez, Victor del Campo, Almudena Rojas, Joaquin Mendoza, Benito Regueiro-Garcia

**Affiliations:** 1Microbiology and Infectology Research Group, Galicia Sur Health Research Institute (IIS Galicia Sur), 36312 Vigo, Spain; carlos.davina@iisgaliciasur.es (C.D.-N.); jorge.julio.cabrera.alvargonzalez@sergas.es (J.J.C.-A.); elenaglezal@gmail.com (E.G.-A.); sergio.silva.bea@usc.es (S.S.-B.); benito.regueiro@usc.es (B.R.-G.); 2Faculty of Biology, Universidade de Vigo, 36310 Vigo, Spain; 3Microbiology Department, Complexo Hospitalario Universitario de Vigo (CHUVI), SERGAS, 36312 Vigo, Spain; miriam.rodriguez.perez@sergas.es (M.R.-P.); maria.del.pilar.figueroa.lamas@sergas.es (M.d.P.F.-L.); 4Virology and Pathogenesis Research Group, Galicia Sur Health Research Institute (IIS Galicia Sur), 36312 Vigo, Spain; alexandre.perez@iisgaliciasur.es; 5Infectious Disease Unit, Internal Medicine Department, Complexo Hospitalario Universitario de Vigo (CHUVI), SERGAS, 36312 Vigo, Spain; 6Preventive Medicine Department, Complexo Hospitalario Universitario de Vigo (CHUVI), SERGAS, 36312 Vigo, Spain; victor.delcampo.perez@sergas.es; 7VIRCELL S.L, Parque Tecnológico de la Salud, 18016 Granada, Spain; arojas@vircell.com (A.R.); jmendoza@vircell.com (J.M.)

**Keywords:** SARS-CoV-2, neutralising antibodies, subgenomic RNA, viral shedding dynamics

## Abstract

The current clinical management of SARS-CoV-2 disease control and immunity may be not optimal anymore. Reverse transcription polymerase chain reaction (RT-PCR) of genomic viral RNA is broadly used for diagnosis, even though the virus may still be detectable when it is already non-infectious. Regarding serology, commercial assays mostly still rely on ancestral spike detection despite significant changes in the genetic sequence of the current circulating variants. We followed a group of 105 non-vaccinated individuals, measuring their viral shedding until negativity and antibody response up to six months. The mean viral detection period until a negative RT-PCR result was 2.2 weeks when using subgenomic RNA-E as a detection target, and 5.2 weeks when using genomic RNA as a detection target. Our neutralising antibody results suggest that, when challenged against a variant different from the variant of first exposure, commercial immunoassays are suboptimal at predicting the neutralising capacity of sera. Additionally, anti-Alpha and anti-Delta antibodies showed very low cross-reactivity between variants. This study provides insights into viral shedding and immune response in pre-Omicron variants like Alpha and Delta, which have been understudied in the published literature. These conclusions point to potential improvements in the clinical management of SARS-CoV-2 cases in order to organise vaccination campaigns and select monoclonal antibody treatments.

## 1. Introduction

Having an accurate knowledge of SARS-CoV-2 viral shedding dynamics and neutralising immunity is key in the control of viral transmission and vaccination campaign organisation. However, despite all the findings about SARS-CoV-2 dynamics since the discovery of this pathogen, there are still insights to be explored.

Reverse transcription polymerase chain reaction (RT-PCR) for genomic viral RNA from a nasopharyngeal swab remains the gold standard for SARS-CoV-2 diagnosis [1]. It must be noted that, in this clinical specimen, viral load in a sample can be more or less representative of the infectious state of the patient, as variations in the swabbing technique could affect the result [2]. Additionally, these techniques cannot distinguish between infectious and non-infectious RNA [3]. This is relevant, as it has been previously established that the period of detectable viral RNA is longer than the period of infectivity [4]. Relying only on RT-PCR data could lead to unnecessary extended quarantines, putting excessive pressure on healthcare systems and in individuals undergoing self-isolation. Nevertheless, while individual viral load measurements may have limitations, the overall quantification of SARS-CoV-2 genomic RNA within a population has shown promise as an indicator of increased viral transmission and, consequently, an early warning signal for potential COVID-19 waves [5].

Viral culture allows for distinguishing infectious from non-infectious samples, but it is unrealistic to establish culture as a routine diagnostic technique as it is more expensive and requires a BSL-3 laboratory to grow the virus. As a second limitation, when facing a new virus, optimal culture conditions are not known, which could limit the accuracy of viral culture as a measurement of viral infectivity at the early stages of facing a new pathogen [6]. Finding a good biomarker for viral infectivity is challenging as the infectious period may be affected by disease severity [7,8], variant [3], age [9], gender [10], or the assay used [11].

Alternative biomarkers of active infection have been tested, such as the detection of subgenomic RNAs (sgRNAs). Since these molecules are necessary for viral protein translation and are generated only during viral replication, they should be only present during periods of active infection [12]. However, more data are required to understand the potential role of sgRNA RT-PCR as a marker of active infection.

An alternative tool for viral RNA quantification is digital PCR (dPCR). Since the beginning of the SARS-CoV-2 pandemic, dPCR has been proposed as a method for improved detection in low viral load samples [13]. Additionally, dPCR could potentially work reliably even without an RNA extraction process [14], which would decrease processing sample time and improve reproducibility.

The gold standard for measuring humoral immunity in vitro is the neutralising antibody assay, and it has provided the prediction of protection since the beginning of the SARS-CoV-2 pandemic [15]. However, these assays are challenging to perform in the routine clinical surveillance setting due to limitations in cost, expertise, and the biosafety levels required to operate live viruses [16,17].

As an alternative, in vitro serological tests such as ELISA and CLIA (enzyme-linked immunosorbent assays and chemiluminescent immunoassays) are cheap, high-throughput, and can provide models to help understand the immunity in big population groups [18]. These tests could also reduce the demands of large trials in vaccine development by using antibody values to measure protection levels (immuno-bridging) [19]. However, serological tests require standardisation [20,21] and cannot differentiate the effectiveness of different antibody fractions against different variants. More notably, most ELISA and CLIA tests for antibody measurements were generated from ancestral antigens, and it has been suggested that their correlation with neutralising capacity is not as solid for Omicron variants as it was for ancestral variants [22]. This effect has been understudied in antibodies generated from other pre-Omicron variants such as Alpha and Delta. Neutralising antibody (nAb) assays can achieve a more accurate measurement of protection, especially against new variants of exposure.

Since the beginning of the SARS-CoV-2 pandemic, vaccination campaigns have been organised against the ancestral variants BA.5 and XBB.1.5 [23]. In 2024, the recommended variant for vaccination was JN.1 or its descendant KP.2 [24,25]. Early vaccines against the Wuhan variant showed a decreased neutralising antibody response against Delta and Omicron variants [26,27], therefore prompting the development of vaccines adapted to the new variants. While it is important to study the impact of booster vaccinations in overall and variant-specific immunity, it is a challenging process. A growing number of the population has a hybrid immunity, with antibody responses generated from subsequent infection and vaccination events, and against different SARS-CoV-2 variants of exposure. Therefore, quantifying the impact of one isolated vaccination event may be limited by the response generated by previous exposures [28].

As most studies have been focused only on hospitalised patients [4], mild COVID has been understudied in comparison, despite being the majority of the infected individuals in a community. This study intends to shed light on the viral dynamics of the pre-Omicron SARS-CoV-2 variants of concern (VOCs), mild disease in unvaccinated individuals, as well as the evolution of their post-infection immune response to understand the role of the specific variant of first exposure (mainly Alpha or Delta) on their humoral response. Additionally, this study will provide insights into potential improvements in the clinical diagnosis of SARS-CoV-2 infection and the accurate estimation of the immunity in the population.

## 2. Results

A total of 105 patients were recruited for follow-up in either viral excretion dynamics (group 1: 73 Alpha, 10 Delta, 22 others) or IgG levels after the infection (group 2: 67 Alpha, 9 Delta, 19 others). The description of the population can be found in Table 1.

### 2.1. Correlation Between RT-PCR Results and dPCR Was Solid Below Cycle Threshold (Ct) 30

All samples with a positive RT-PCR result from RNA extracts in the study were quantified using QiaAcuity digital PCR (Qiagen, Hilden, Germany) and crude samples as inputs (direct quantification). Prior to the assays, direct quantification by dPCR was compared to RNA extracts to confirm comparable results (Appendix A), as it has been previously analysed in another study [14].

Results of dPCR showed a high correlation with RT-PCR-positive samples (R^2^ = 0,84; Figure 1a). This correlation was high for a cycle threshold (Ct) value below 30 (R^2^ = 0.87) (Figure 1b), and negligible in samples with a Ct value over 30 (R^2^ = 0.08), showing high uncertainty in viral load determination in samples of a low viral load. According to the linear model (Figure 1b), a Ct of 30 was equivalent to 6–7 molecules per microliter in the dPCR data.

### 2.2. Convalescent Period with Low Viral Load Only Detected by Genomic RNA RT-PCR

In order to get a more realistic grip on infection dynamics, all samples were ordered by the day with the detection of maximum viral load. This allowed us to normalise all patients according to their infection dynamics, with samples from patients still in the early phases of infection (detection before maximum viral load) and samples from patients in late phases (detection after maximum viral load). There was a period of 20 days with results above 10 genomes/uL, showing a bell curve graph (Figure 1c). After the second week post-infection, a second period was found where RNA was still detectable in most patients but the viral load was residual, from an average of 10^3.73^ genomes/μL at the time of diagnosis to 10^1.75^ at week 1 and 10^0.56^ at week 2 (Figure 1d). Only seventeen patients (16.2%) showed viral loads below Ct 30 from week 2 onwards, and only four (3.8%) from week 3 onwards. However, in 54.3% of the individuals (57/105), viral RNA was still detectable after week 4 after diagnosis (Figure 1f). These late-time positive samples were of low concentration and, considering the data above, challenging to quantify accurately with both qPCR and dPCR.

### 2.3. sgRNA Was Only Detectable in the First Weeks of Infection in Most Samples

Subgenomic RNA, which has been used in previous studies as a marker of viral infectivity [29], was tested in 97 out of 105 patients for sgRNA. Negative sgRNA samples were concentrated in the latest days of detectable shedding with the lowest RNA viral loads (Figure 1c). When measuring genomic RNA, the mean weeks until a negative test (no positive target) was 5.2 ± 2.4 weeks, while with sgRNA, it was 2.2 ± 1.4 weeks.

As it is the case with the samples with high viral load, a majority of the samples positive in sgRNA occurred in weeks 0–1 after diagnosis. The positivity ratio in sgRNA (amongst positive samples as detected by traditional RT-PCR) was 94.4% (102/108) in the week of diagnosis, 75.8% (66/87) in the following week, and 16.0% (8/50) in the second week, dropping below 10% in the weeks onwards (Figure 1e). Therefore, starting in the second week after diagnosis, 10.5% of the samples with detectable RNA were negative in sgRNA. The number of days after diagnosis was a good predictor of sgRNA positivity, with a positivity of 84.9% (169/199) in the first 10 days, and 10.1% (15/148) in the following days until undetectable gRNA. At week 3, 69.1% (67/97) of patients already had a negative result in sgRNA (Figure 1g).

Notably, one participant tested negative in sgRNA in week 3 and then showed detectable sgRNA again two months after infection, with its genomic RNA Ct value rising to 24.6 in this period. Next-Generation Sequencing (NGS) data confirmed that this participant had the same SARS-CoV-2 strain throughout all the follow-up period, making it unlikely that these were two separate cases of infection.

### 2.4. Age Had a Moderate Impact in Peak Viral Load

Variables with an impact in peak viral load and days of excretion were checked in the population. Age had a mild impact on peak viral load, with older people having a higher viral load (ρ = 0.257, *p* = 0.008; Spearman’s rank correlation coefficient) (Appendix A). The number of excretion days was not significantly affected (ρ = 0.153, *p* = 0.120) (Appendix A). Gender had no impact in excretion dynamics, nor in peak viral load, nor in excretion days (*p* = 0.367; *p* = 0.194, respectively; Wilcoxon signed-rank test) (Appendix A).

### 2.5. IgG Levels Peak One Month After Infection

Sera samples were taken from 95 participants in order to measure variations in the IgG levels after infection in a 6-month period. Given that the follow-up time of this study overlapped with the SARS-CoV-2 vaccination campaign in Spain, it was an opportunity to study the impact of the vaccination after the natural infection. The amount of participants vaccinated at each timepoint is described in Table 2.

Unvaccinated individuals throughout the whole follow-up period had a peak of antibody levels at month 1. Although there was a drop of antibody count at months 3 and 6, the difference was non-significant (Figure 2, Appendix A, Dunn’s test).

Participants showed a boost in antibody levels in the sample immediately after vaccination, with month 3 showing the highest levels measured in the patients with their vaccine taken immediately between months one and three (Figure 2). Nonetheless, at month 6, all patients returned to antibody counts with no significant difference with the antibody values at month 1 (pre-vaccination) (*p* = 0.187, Appendix A). Participants receiving a vaccine dose between months 3 and 6 also showed an increase of antibody levels at month 6, after a non-significant drop between months 1 and 3 (*p* = 0.187, Appendix A).

### 2.6. Stronger Neutralisation Capacity Against the Variant of Infection

Samples from one month after diagnosis were selected for neutralisation assays, as in this group there were no vaccinated individuals. Individuals were selected from those infected with Alpha and Delta variants, the two most common variants at the time of recruitment.

Nine participants were infected with the Delta variant. In order to compare anti-Alpha and anti-Delta sera, 18 samples were selected for the pseudovirus neutralisation assay. In addition to the nine patients with anti-Delta sera fractions, nine participants with anti-Alpha sera were tested. Samples were selected in order to have paired antibody values between Alpha and Delta as measured by an in vitro immunoassay (Figure 3a). Neutralisation was performed against a Delta-spike carrying pseudovirus and measured as 50% inhibitory dilution (ID50).

Despite paired antibody values measured in vitro (Figure 3a, *p* = 0.73), antibodies generated by individuals infected with the Alpha variant (anti-Alpha) performed significantly worse against the Delta variant lentiviral vector than the antibodies generated against the Delta variant (anti-Delta) (log2 ID50: 8.72 ± 1.57 (alpha) vs. 10.88 ± 1.89 (delta); *p* = 0.024) (Figure 3b). Notably, the prediction of Delta neutralisation capacity by the in vitro assay (CLIA) was better for anti-Delta antibodies than for anti-Alpha, with anti-Alpha CLIA results showing no significant correlation with the Delta neutralisation capacity (anti-Alpha ρ = 0.40, *p* = 0.29; anti-Delta ρ = 0.88, *p* = 0.003; Spearman’s rank correlation coefficient) (Figure 3c).

### 2.7. Live Virus Neutralisation Assay Showed a Lower Omicron Neutralisation Capacity for Sera Generated Against Pre-Omicron Variants of Concern (VOCs)

Neutralisation capacity was also measured using a live virus assay. In addition to the 18 samples tested with the pseudovirus assay, additional samples were added from the available Alpha group, all from 1 month after infection. Sera samples from anti-Alpha (n = 21) and anti-Delta (n = 9) were tested against the viral variants ancestral D614G, Alpha, Delta, BA.2 (Omicron), and XBB.1.5 (Omicron). The neutralisation capacity was measured as the maximum inhibitory sera dilution where the cytopathic effect was still observed. Comparing all variants tested, both sera performed best against their own variant (Alpha and Delta, Figure 3d,e). The anti-Alpha sera performed significantly better against Alpha than against all other variants tested, while the anti-Delta sera performed significantly better against Delta than against both Omicron variants tested (Dunn’s test, Appendix A).

## 3. Discussion

Our data suggest a long convalescent period where RNA is still detectable, with a mean of 5.2 ± 2.37 weeks. This is in contrast to other publications, where the median of viral RNA shedding was around 17 days [4]. This is likely due to methodological differences. In the case of our study, no patient was considered negative until all three targets were negative, with a maximum Ct value of 40. Other studies on viral shedding have used various targets and tests and used Ct 40 [30], Ct 38 [9], or Ct 33 as thresholds [31] amongst others, with varied results. Notably, our study measured one sample per week, as per the normal testing period in mild individuals during their isolation period in Spain. Daily testing, as performed in multiple studies with hospitalised patients, allowed for the faster detection of negativity due to a higher number of datapoints. Isolation periods and patient release can be optimised with increased testing, but this can be challenging for healthcare services working with mild patients isolated in their own place of residence.

The duration of infectious virus shedding was also found to be affected by variables such as age, gender, or viral variant. In our study, older participants had a higher viral load, even being individuals with mild disease (*p* = 0.008). There was no difference in shedding duration (*p* = 0.120). However, our study focused on adults between 18 and 55 years old. Elderly participants were discarded from inclusion as they were entering the vaccination campaign during the time of recruitment. A different study found faster viral clearance in younger participants, but only in individuals under 18 [9]. Additionally, males showed slower shedding than females in severe cases [10]. Our data, in mild patients, showed no difference in viral shedding by sex (*p* = 0.194). These variations highlight the complexities to determine, without running culture tests, when a patient can be released from confinement without the risk of the expansion of the virus.

In order to quantify samples in a more effective manner, our samples were also measured by dPCR. dPCR can more accurately quantify viral RNA than qPCR, as it can provide absolute quantification by partitioning the RT-PCR reaction mixture in nanowells containing, ideally, 0 or 1 target molecules, improving accuracy compared to qPCR [32]. From a clinical perspective, it provides the advantage of working without the need of RNA extraction, as partitioning the sample would also partition inhibitors present in the sample [32,33]. Avoiding the RNA extraction step could allow for fast diagnosis in cases of an emergency. Our data also supported the use of direct quantification for dPCR, as we found high correlations in the results between direct quantification and dPCR on an RNA extract (Appendix A).

The correlation between dPCR and RT-PCR was high (R^2^ = 0.87), except for samples of low viral load, below the estimated limit of quantification for RT-PCR and dPCR [34,35]. Although dPCR can improve detection when the target concentration is low (above Ct 30) [13], both techniques show high quantification uncertainty in such cases, mainly due to undersampling errors [36].

Our data showed that, in mild COVID, after week one, the viral load was mostly residual, and probably associated with the shedding of non-infectious viral RNA [37]. Therefore, late Ct-positive results could be irrelevant in the context of isolation measurements, as Ct values correlate inversely with the probability of infectiousness [38]. We measured sgRNA in an additional RT-PCR assay, as it has been proposed as a good biomarker of active infection [7,29,35,39,40]. Nevertheless, it must be noted that another publication found sgRNA to have a poor yield as a measure of infective virus [41]. It has also been suggested that the decay ratio of sgRNAs after the inhibition of transcription is not fast enough for sgRNAs to be an accurate predictor of infectivity [42].

In our cohort, a majority of individuals showed their first negative sgRNA result between week 1 and week 2, which is a timeframe similar to that expected for infectivity [41,43]. This shows the likelihood of sgRNA being a more accurate measurement of active infection than genomic RNA, and therefore, it could be implemented as a diagnostic tool to avoid increasing isolation periods or hospital bed occupation. Notably, some studies have shown that while the presence of sgRNA may not indicate infectiousness, probably due to the stability of the molecules in clinical samples, the absence of sgRNA would indicate the absence of infectivity. Therefore, sgRNA has a high sensitivity as a marker of infectiousness despite its specificity being suboptimal [29,41]. In this model, a negative RT-PCR could be enough to consider an individual non-infectious.

Our study showed a strong and variant-specific neutralising immune response after mild SARS-CoV-2 infection. There was an increase of the IgG response after one dose of vaccine after the infection as measured by CLIA. Similarly, past studies have shown that hybrid immunity, a combination of infection + vaccine, provides a higher IgG response than a regular vaccination pattern with two doses (vaccine + vaccine) [44,45,46].

The duration of the IgG response after vaccination is still a relevant issue during the SARS-CoV-2 vaccination campaigns, as the half-life of antibodies in blood is important for vaccination booster scheduling. One study showed a 50% decline in antibodies in a 1–3 month period after the second dose, even in the case of pre-exposure to infection [47]. Similarly, in this study, post-infected patients with one dose of vaccine showed a mean decrease of 53% of IgG levels three months after immunisation (22.4 ± 12.07 in month 3; 10.4 ± 6.71 in month 6 (mean ± IQR)). Notably, the IgG levels in month 6 showed comparable levels to the IgG levels prior to vaccination (*p* = 0.187). This steep decrease did not happen in the post-infection sera.

IgG levels are not the only relevant measure in humoral immunity, as different antibody fractions have different neutralisation capacities against different variants. It has been suggested that up to 66% of antibodies from a convalescent response are not cross-reactive and are only able to neutralise one variant [48]. While the immunoassays (ELISA, CLIA) will only give one readout as an output, the neutralising antibody assay can be used to challenge sera against different viral variants. Recent reports have suggested that immunoassays are not accurate when measuring antibodies against different clades. Firstly, some immunoassays have been associated with reduced sensitivity to antibodies generated after Omicron infection, due to the antigens used in these kits being not updated to current circulating variants [49,50]. More recently, the correlation between ELISA results and neutralising antibodies was found to be below 0.30 (Pearson’s coefficient) when measuring neutralisation against the Omicron variants BQ and XBB [22].

Accurately quantifying immune boosting has been especially relevant in the last vaccination campaigns with the bivalent vaccines Wuhan/BA.5 (2022 campaign) and the monovalent XBB.1.5 (2023 campaign). This is because some publications have suggested that after booster vaccination with Omicron, the memory B cell response is directed towards epitopes of previous variants, decreasing the effect of boosting against new circulating variants (immune imprinting) [51,52,53]. The majority of individuals in the population have had their first exposure to SARS-CoV-2 in the Wuhan D614G variant, either due to the first available vaccines targeting against this variant or due to infection in the period before the variants of concern arose. Because of this, many publications have shown that neutralising capacity is highest against the ancestral variants than against other exposure variants, even after bivalent vaccination [54,55,56], omicron breakthrough infection [54,55], or monovalent XBB.1.5 vaccination [57]. A recent publication also showed that sera taken from individuals in 2023 had a higher neutralisation of the ancestral variant than sera taken in 2020 after infection with the same variant [22]. In all cases, the variant of first exposure had the highest neutralising capacity. Our data suggest that this is not only true in the ancestral variant, but also in Alpha or Delta. Sera from vaccine-naïve participants infected with the Alpha and Delta variants (anti-Alpha and anti-Delta sera) were used for our neutralisation experiments. Individuals with anti-alpha sera showed the highest neutralization against the Alpha variant, and the anti-Delta sera showed the highest neutralisation against the Delta variant, even higher than that against the Wuhan D614G variant (Figure 3d,e). The data suggest that most individuals have the highest neutralisation against their variant of first exposure, regardless of the variant. Vaccine-naïve individuals to SARS-CoV-2, such as infants receiving their first exposure to current variants, could be more effective at neutralising these variants than individuals who were previously exposed to pre-Omicron variants, despite boosters.

In our data, regarding the Omicron variant landscape, neutralisation was significantly lower against Omicron variants than against the variant of exposure, including in XBB.1.5, the vaccine variant of last season (Figure 3d,e). It is challenging to compare neutralisation against BA.2 and XBB.1.5 in our samples, as in both cases, most serum samples fell below the limit of the detection of neutralisation. Regardless, our data show that antibodies targeting Alpha and Delta epitopes, perhaps more uncommon in the population as there have not been vaccination campaigns targeting these variants, show little neutralisation capacity against Omicron variants, at least after one exposure. A recent study about the polyclonal responses from convalescent individuals found no nAbs with cross-reactivity between Delta and Omicron, which could explain the poor performance of anti-Delta sera against the Omicron variants tested in our study [48]. Neutralising Abs for one SARS-CoV-2 variant can even be enhancing for other variants, at least in vitro [58].

During the COVID-19 pandemic, CLIA results have been used in the clinical context, such as in the detection of hyperimmune sera for its use as passive immunity [59]. If CLIA results are to be accurately used in the clinic, there is a need to obtain high-throughput in vitro assays with reagents that can be variant-specific in order to better estimate immunity to a specific variant. This is even more relevant in the current context of SARS-CoV-2, where most individuals in the population have been exposed to various variants via both natural infection and vaccination.

This study has the following limitations: the infectivity of clinical samples was not measured in cell cultures, and the neutralisation sample number was low due to the low variability of variant circulation at the time of recruitment (69.5% patients infected with the Alpha variant). However, the study presents a significant difference between the detection period with genomic RNA and subgenomic RNA. This is relevant as subgenomic RNA has shown accuracy as a proxy for a viable virus in previous studies [29].

Our study highlights the variation in viral detection using different RNA targets, showing the relevance of methodology in viral shedding studies. Determining accurately the moment when the virus is not viable anymore and an individual stops being infectious is key, as it could help guide future clinical strategies regarding when to release individuals from isolation. It could also be useful in detecting when an individual is free of disease in order to start receiving immunosuppressive therapies. Additionally, the study found a high increase in IgG response after hybrid immunity (infection boosted with one dose of vaccination), although this increase in IgG is short-lived. Our data also suggest that neutralisation capacity is highest against the variant to which the immune system was first exposed, regardless of future expositions via immunisation or vaccination. Finally, the low accuracy of commercial immunoassays in the prediction of neutralising capacity against new incoming variants suggests the need for updating reagents. These conclusions could guide microbiological diagnosis, clinical response, and vaccination schemes in future times of healthcare crisis.

## 4. Materials and Methods

### 4.1. Patient Selection and Sample Collection

A total of 105 patients diagnosed with infection by SARS-CoV-2 were recruited for the study. Ten participants dropped out of the study during the follow-up (Figure 4).

For the measurement of viral dynamics, one nasopharyngeal sample was taken per week until the virus was undetectable (negative RT-PCR for all three targets). The samples were collected in Vircell transport medium (Vircell, Granada, Spain). For the study of the immune response dynamics, a serum sample was taken from patients at months 0, 1, 3, and 6 after infection in order to measure antibody levels. As participants were non-hospitalised and quarantined in their places of residence, a healthcare professional was in charge of visiting participants and obtaining the samples until they could go to the hospital for the sample collection.

The study received approval from the Galician Network of Research Ethics Committees (Spain) (protocol number 2020/627), adhering to the principles outlined in the Declaration of Helsinki. All methodologies were conducted in accordance with relevant guidelines and regulations, and all participants provided informed consent prior to their inclusion in the study.

### 4.2. Inclusion Criteria

The inclusion criteria included non-hospitalised patients aged 18–55 years, with laboratory-confirmed SARS-CoV-2 infection by RT-PCR in the recruitment period (March–July 2021). All participants signed an informed consent form.

### 4.3. Viral RNA Quantification

#### 4.3.1. RT-PCR

Nucleic acid extraction was performed on a Microlab STARlet IVD platform (HAMILTON) using a STARMag 96 × 4 Universal Cartridge Kit (Seegene Inc., Seoul, Republic of Korea). RT-PCR was performed with an Allplex 2019-nCoV Assay, (Seegene Inc.) and a Bio-Rad CFX96 Touch™ (Bio-Rad Laboratories, Inc., Hercules, CA, USA), targeting the Envelope (E), Nucleocapsid (N), and RNA-Dependent RNA Polymerase (RDRP). An additional RT-PCR test was performed to target the E-gene sgRNA, using the primers and probe previously published by Wölfel et al. [7] and a Lightcycler Multiplex RNA Virus Master (Roche Diagnostics, Mannheim, Germany) on a z480 thermocycler (Roche Diagnostics), with a final concentration of primers of 0.1 μM and a probe of 0.05 μM.

#### 4.3.2. Digital PCR (dPCR)

Nasopharyngeal exudate crude samples were quantified (direct quantification) using a QiAcuity One-Step Viral RT-PCR kit from Qiagen (Qiagen, Hilden, Germany) with a 26k nanoplate following manufacturer’s specifications. The primer/probe used was the COVID-19 E-gene primer-probe reagent (TIBMOLBIOL, Berlin, Germany) with 1 μL per 40 μL of reaction.

Following recommendations from dMIQE (Minimum Information for Publication of Digital PCR Experiments) [36], only samples with over 15,000 valid partitions and below a 0.9 positive partition ratio were considered as valid. In order to avoid nanoplate saturation, samples were diluted based on the Ct value obtained from RT-PCR as follows: Ct < 12 dilution 1:1000, Ct = 12–14 dilution 1:200, Ct = 14–15.5 dilution 1:100, Ct = 15.5–16 dilution 1:50, Ct = 16.5–18 dilution 1:20, and Ct 18–19 dilution 1:10.

### 4.4. SARS-CoV-2 Next-Generation Sequencing (NGS)

One sample per patient was selected for viral whole genome sequencing by NGS. For NGS, 11 μL of extracted RNA was subjected to retrotranscription using a SuperScript IV kit (SSIV, Invitrogen by Life Technologies, Carlsbad, CA, USA) according to manufacturer’s specifications and using random hexamers as primers. cDNA was then amplified using Q5 High Fidelity DNA Polymerase (New England Biolabs, Ipswich, MA, USA) and the ARTIC v3 primer panel. Enriched samples were then normalised, and libraries were prepared using an Illumina DNA prep kit (Illumina Inc., San Diego, CA, USA) using 1/4 of the recommended volume. The Illumina iSeq platform was used for sequencing. The fastqs generated were analysed as described in https://github.com/OMIC-G/COV (accessed on 19 March 2025).

### 4.5. Measurement of Antibodies in Sera

Sera samples were taken at months 0, 1, 3, and 6 after diagnosis and analysed using the COVID-19 VIRCLIA IgG Monotest (Vircell Microbiologists, Granada, Spain), a CLIA in vitro assay targeting IgG antibodies against the receptor binding domain of the spike protein.

### 4.6. Pseudovirus Neutralisation Assay

The protocol for the pseudovirus neutralisation assay was based on that previously published by Nie et al. [60]. Commercially available lentiviral vectors transfected with green fluorescent protein (GFP) genes and presenting the SARS-CoV-2 spike protein variant Delta (B.1.617.2) were used (VectorBuilder, Neu-Isenburg, Germany). Sera was heat-inactivated (30 min, 56 °C) and serially diluted (2×) in duplicates five times. The lentiviral vectors were added to the sera in a virus/cell ratio (Multiplicity of Infection, MOI) of 50. A positive control (no serum, viral control) and a blank (no virus, cellular control) were added on each plate in triplicate.

Incubation in 96-tissue culture-treated well plates was performed for 1 h at 37 °C. Then, HEK293T-ACE2 cells (25,000 cells/well) were added and incubated for 72 h at 37 °C, 5% CO_2_ in a FLUOstar Omega Fluorescence plate reader with OMEGA control software version 5.70 R2 (BMG LABTECH, Offenburg, Germany). After 72 h, fluorescence was measured as the sera dilution that could reduce fluorescence to 50% of the viral control (no sera) (50% inhibitory dilution (ID50)). ID50 was calculated using the Reed–Muench method.

### 4.7. Live Virus Neutralisation Assay

For the live virus neutralisation assay, heat-inactivated sera (30 min, 56 °C) were serially diluted (2×) eight times starting at 1:20 dilution, and then incubated for 1 h at 37 °C with viral samples diluted to a viral infectivity of 100 times the median tissue culture infective dose (100× TCID50).

The viral variants used for neutralisation were ancestral (D614G), Alpha (B.1.1.7), Delta (B.1.617.2), Omicron (BA.2), and Omicron (XBB.1.5). After incubation, the mix was added to a 96-well plate seeded with 25,000 Vero cells/well. After five days of infection, the maximum inhibitory concentration was calculated as the maximum serum dilution with no observed cytopathic effect. An infection control (no serum) and a positive control (high-titre serum) were added in each experiment.

### 4.8. Data Analysis

All statistical data analyses were performed using R (version 4.1.1, https://cran.r-project.org/, accessed on 24 March 2025). Data visualisation was performed with the R program ggplot2 [61]. In all boxplots generated, the middle line is the median, the upper and lower limit of the hinges correspond to the third and first quartile, respectively, and the whiskers correspond to the largest and smallest values no further away than 1.5 × IQR (Interquartile Range) [62]. A Shapiro–Wilk normality test was performed to check for normality. Kruskal–Wallis and Dunn’s tests were used when indicated. When multiple pairwise tests were performed simultaneously, the p-value was adjusted using the Holm correction. Pearson’s correlation coefficient (R^2^) and Spearman’s rank correlation coefficient (ρ) were used for rank correlations between two variables when indicated.

## Figures and Tables

**Figure 1 ijms-26-02948-f001:**
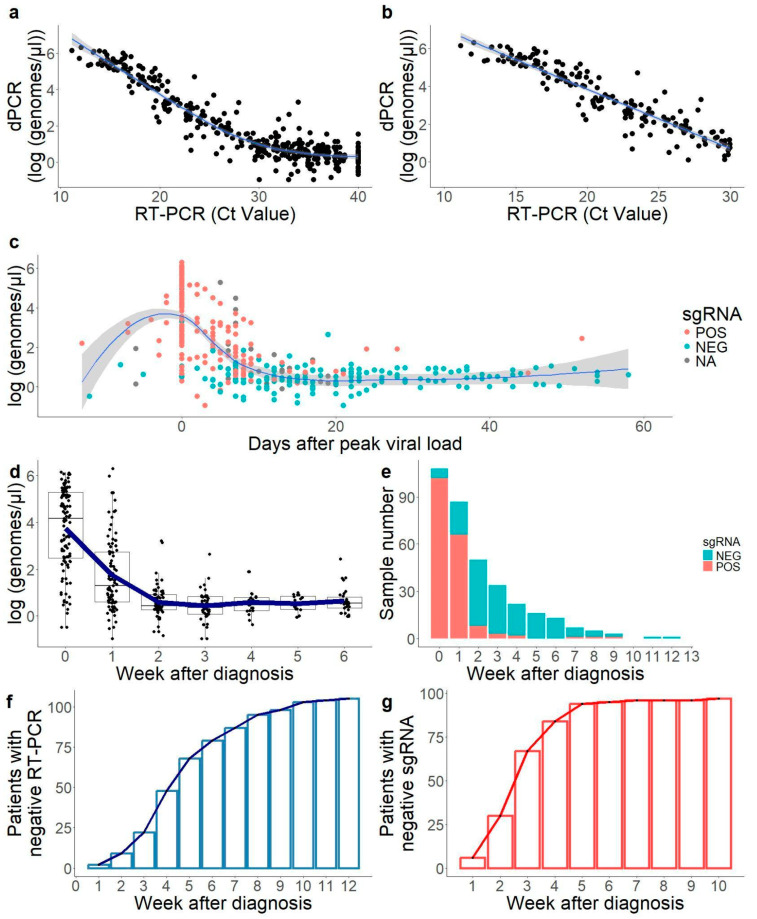
Study of SARS-CoV-2 viral excretion dynamics. (**a,b**) Correlation of RT-PCR (Ct value) and dPCR (absolute quantification) in the estimation of viral load. Correlation was linear until Ct = 30, as shown on the panel on the right (Pearson correlation coefficient in samples below Ct 30: 0.87; in samples above Ct 30: 0.08). Blue lines: LOESS (panel (**a**)) and linear (panel (**b**)) approximation model. (**c**) Excretion dynamics from all positive samples in the study. Day 0 is marked for all subjects as the day of the PCR with the highest viral load. Blue line: LOESS approximation with grey area showing the 95% confidence interval. Point colour indicates the detection of subgenomic RNA (sgRNA) in each sample, with NA indicating samples that were not tested for sgRNA. (**d**) Boxplot of viral load of all samples per week after diagnosis. The blue line indicates the mean value per week. Six (6) refers to six or more weeks in this plot. (**e**) Amount of samples positive in gRNA that were positive in sgRNA (red) or negative (blue) per week after diagnostic. (**f**) Amount of patients with a negative RT-PCR in gRNA per week. (**g**) Amount of patients with a negative RT-PCR in sgRNA per week. All data used to generate this figure are available in Appendix A.

**Figure 2 ijms-26-02948-f002:**
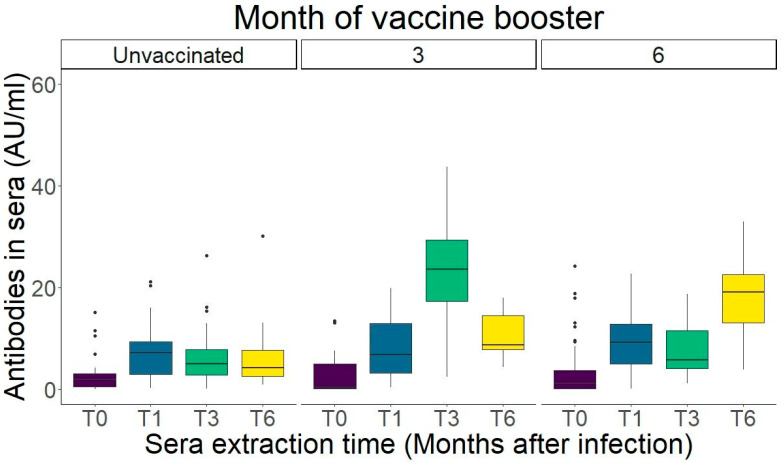
IgG levels of patients according to vaccination time. (**Left**) Unvaccinated individuals. After a peak in month 1, antibody AUs decreased moderately in months 3 and 6. (**Middle**) Individuals vaccinated before month 3. A boost of IgG from vaccination is shown. However, at month 6, antibody levels already dropped to around pre-vaccination levels (month 1). (**Right**) Individuals vaccinated before month 6. Antibody boost caused by vaccination at the month 6 measurement. Dots indicate datapoints above 1.5 times the interquartile range (IQR). All data used to generate this figure are available in Appendix A.

**Figure 3 ijms-26-02948-f003:**
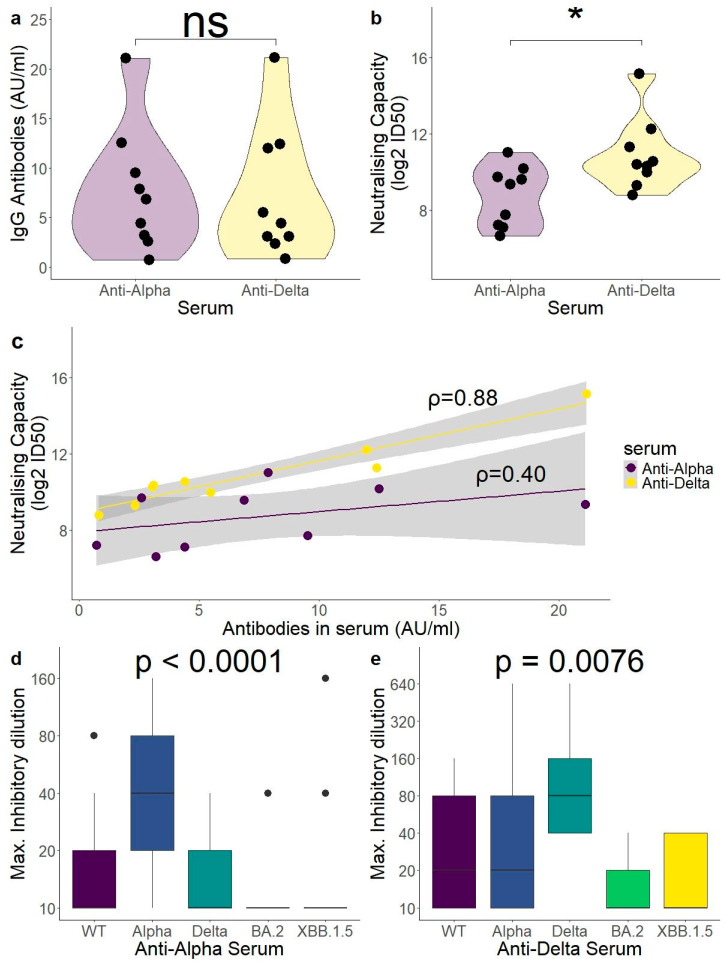
Neutralisation capacity post-infection was highest against variants of infection. (**a**) A total of 18 sera samples were selected for pseudovirus neutralisation, 9 from patients infected with the Alpha variant and 9 infected with the Delta variant, with paired IgG CLIA results (*p* = 0.73). (**b**) Neutralising capacity of Delta-spike carrying pseudovirus particles. Anti-Delta sera were more successful at neutralising the Delta pseudovirus than anti-Alpha sera (*p* = 0.024). (**c**) The CLIA assay was better at predicting Delta neutralisation capacity in anti-Delta than in anti-Alpha sera (*p* = 0.003 vs *p* = 0.29). (**d**,**e**) Live virus neutralisation assay showed the best neutralisation capacity for variants of infection and a significant drop in the neutralisation of Omicron variants. Minimum sera dilution tested was 1:20, with a value of 10 being indicated for samples with no neutralisation detected. For (**a**,**b**), the Wilcoxon signed-rank test. For (**c**), Spearman’s rank correlation coefficient. The lines represent the linear model approximation with the grey area being the 95% confidence interval. For (**d**,**e**), Kruskal–Wallis test. All data used to generate this figure are available in Appendix A. ns = non significant. * = *p* < 0.05.

**Figure 4 ijms-26-02948-f004:**
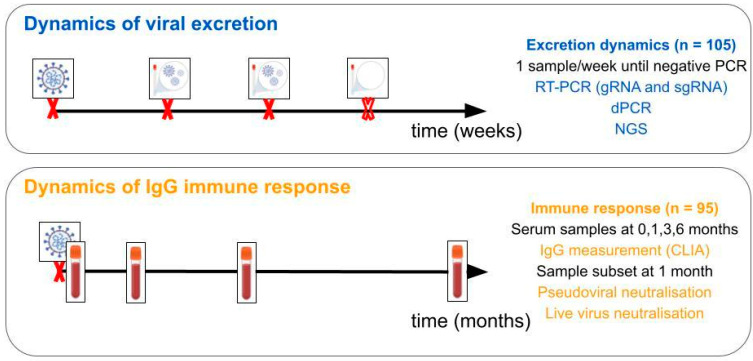
Graphical summary. For the first aim of the study, viral load was determined in one sample per participant and week until negative on three targets of an RT-PCR test. For the second, serum samples were taken at 0, 1, 3, and 6 months after infection in order to measure IgG levels and study neutralisation capacity.

**Table 1 ijms-26-02948-t001:** Description of the population participating in the study. Group 1: excretion dynamics. Group 2: humoral immunity.

	Group 1 (*n* = 105)	Group 2 (*n* = 95)
**Age (mean ± sd)**	37.0 ± 9.46	38.0 ± 8.87
**Gender**		
Female	61 (58.1%)	58 (61.1%)
Male	44 (41.9%)	37 (38.9%)
**SARS-CoV-2 variant**		
Alpha	73 (69.5%)	67 (70.5%)
Beta	2 (1.90%)	2 (2.11%)
Gamma	5 (4.76%)	4 (4.21%)
Delta	10 (9.52%)	9 (9.47%)
Mu	6 (5.71%)	4 (4.21%)
No sequence	9 (8.57%)	9 (9.47%)

**Table 2 ijms-26-02948-t002:** Individuals vaccinated across the follow-up vigilance period. All subjects who were vaccinated had only one dose after infection.

Participants	Month 0	Month 1	Month 3	Month 6
Unvaccinated	88	94	76	18
Vaccinated	0	0	19	58

## Data Availability

The original contributions presented in this study are included in the article/Appendix A. Further inquiries can be directed to the corresponding author.

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
