# Peer review of "Subgenomic RNA and Limited Cross-Reactive Neutralising Antibodies Point to Potential Improvements in SARS-CoV-2 Clinical Handling"

_ijms, 2025, doi:10.3390/ijms26072948_

Round 1
Reviewer 1 Report
Comments and Suggestions for Authors
This study examined viral shedding dynamics and antibody responses in unvaccinated individuals infected with pre-Omicron SARS-CoV-2 variants, predominantly Alpha and Delta. Comparing genomic and subgenomic RNA detection, the authors demonstrated that sgRNA probably more accurately reflects active infection and exhibits shorter shedding durations. Furthermore, the study assessed neutralizing antibody cross-reactivity, revealing limited efficacy against divergent variants. These findings contribute to improved understanding of SARS-CoV-2 infection and inform the development of enhanced clinical management and diagnostic strategies. Overall, the study employs a robust experimental design and performed a straightforward analysis of viral dynamics and antibody responses in pre-Omicron SARS-CoV-2 infections. Although the conclusions may not present entirely novel findings, they effectively validate expected patterns in patient cohorts. I recommend that the authors carefully review their figures for accuracy, and specifically clarifying the representation of the bars (mean or median) and error bars (standard deviation or standard error of the mean).
Reviewer 2 Report
Comments and Suggestions for Authors
In general, the manuscript entitled “Subgenomic RNA and limited cross-reactive neutralising anti-bodies point out to potential improvements in SARS-CoV-2 clinical handling” is a well-draft manuscript. Still, the manuscript needs some corrections and editions as follows:
- The plagiarism rate is extremely high: 78% overall, with 62% coming from a single study. Please fix this issue.
Abstract
- Line 22. “RT-PCR”. Write the full name followed by the abbreviation in parentheses.
- L27-28. Rewrite to clarify the meaning.
Keywords
- L 35. Replace “sgRNA” With “subgenomic RNA ”
Introduction
- L 43. Add a reference.
- “RT-PCR”. Write the full name followed by the abbreviation in parentheses.
- L 74. “in vitro”. Should be in italics, apply this comment throughout the manuscript.
- L 84. “ELISA and CLIA”. Write the full name followed by the abbreviation in parentheses.
- L 102. Rewrite to clarify the meaning.
Results
- L 116. “Qiagen”. Add country.
- L 122. “Ct”. Write the full name followed by the abbreviation in parentheses.
- L 171. “ NGS”. Write the full name followed by the abbreviation in parentheses.
- L 177. “p”. Should be in italics, apply this comment throughout the manuscript.
Discussion
- L 317. Rewrite to clarify the meaning.
- L 355. Rewrite to clarify the meaning.
- L 377. Add conclusion section/paragraph.
Materials and Methods
- L 405-406. Add country name.
Reviewer 3 Report
Comments and Suggestions for Authors
I am very grateful for the opportunity to read this manuscript. It clearly shows that PCR of genomic viral RNA collected from nasopharyngeal swabs is not the gold standard, and I really hope that this manuscript will be published soon. Researchers around the world should know that the detection period by PCR is too long. Yes, it is unnatural that we still need BSL3 facilities to culture SARS-CoV-2, and there are so many inconveniences in research under the current circumstances.
I think that the experimental methods and results in this manuscript are sound.
It is also easy to understand that the immunoassay method for antibody detection is not always accurate.
My concern is the description of the limitation on line 378. "Nonetheless, the study shows that demonstrating the absence of sgRNA in clinical samples should guide future strategies in releasing critical individuals from isolation or even to receive immunosuppressive therapies." Could you please explain this part in more detail? That is the only thing that bothers me.
Round 2
Reviewer 2 Report
Comments and Suggestions for Authors
The authors have done the requested changes.
Author Response
Comment:
The authors have done the requested changes.
We appreciate the comment. We believe that the suggestions presented by the reviewer have improved the overall quality of the manuscript and therefore we thank him for his work.